# Simulation Analysis of Thermoacoustic Effect of CNT Film with Metasurface-Enhanced Acoustic Autofocusing

**DOI:** 10.3390/nano14181481

**Published:** 2024-09-11

**Authors:** Dalun Rong, Zhe Li, Qianshou Qi, Zhengnan Liu, Zhenhuan Zhou, Xinsheng Xu

**Affiliations:** 1School of Aeronautics and Astronautics, Shenzhen Campus of Sun Yat-sen University, Shenzhen 518107, China; rongdlun@mail.sysu.edu.cn; 2School of Civil Engineering, Hunan University of Technology, Zhuzhou 412007, China; 3State Key Laboratory of Structural Analysis, Optimization and CAE Software for Industrial Equipment, School of Mechanics and Aerospace Engineering, Dalian University of Technology, Dalian 116024, China; zheli20242024@gmail.com (Z.L.); qiqs@mail.dlut.edu.cn (Q.Q.); zhouzh@dlut.edu.cn (Z.Z.); 4School of Traffic and Transport Engineering, Changsha University of Science & Technology, Changsha 410114, China; lzn@csust.edu.cn; 5Hunan Communications Research Institute Co., Ltd., Changsha 410015, China

**Keywords:** thermoacoustic effect, acoustic metasurfaces, Airy beams, acoustic focusing

## Abstract

This study introduces a novel thermoacoustic (TA) focusing system enhanced by Airy beam-based acoustic metasurfaces, significantly improving acoustic focusing and efficiency. The system integrates a TA emitter, fabricated from carbon nanotube (CNT) films, with a binary acoustic metasurface capable of generating quasi-Airy beams. Through finite element simulations, the system’s heat conduction, acoustic focusing, and self-healing properties were thoroughly analyzed. The results demonstrate that the system achieves superior sub-wavelength focusing, tunable focal length via frequency control, and robust self-healing, even in the presence of obstacles. These findings address current limitations in TA emitters and suggest broader applications in medical ultrasound and advanced technology.

## 1. Introduction

In recent years, nanotechnology made great progress in fields such as materials science, electronics, and medicine. The unique properties of nanomaterials, particularly their large surface area and enhanced physical characteristics, became a focal point in various technological applications. One notable application is the thermoacoustic (TA) effect, which involves converting thermal energy into acoustic energy. The introduction of nanomaterials, such as carbon nanotube (CNT) films [1], graphene film [2,3], and MXenes [4] enabled significant TA effects that were difficult to achieve with conventional materials [5], owing to their high surface area and low heat capacity. This opened up new possibilities for efficient energy conversion using the TA effect [6,7,8,9].

Consequently, TA emitters attracted considerable attention [10,11,12]. These devices consist of a time-varying thermal source made from nanomaterials, typically activated by electrothermal modulation to generate Joule heating, and a supporting substrate. The thermal source induces heat exchange with the surrounding medium, creating a dynamic spatial temperature gradient that leads to sound emission from the medium. Unlike traditional transducers that rely on diaphragm vibrations for energy conversion, TA emitters convert thermal energy to acoustic energy without vibrations. This allows for a broader frequency response (1 Hz–20 MHz) [13] and a simpler structure, making TA emitters ideal for applications in wearable medical devices, such as artificial throats [14,15,16], and wearable ultrasound patches [17], potentially useful in ultrasonic therapy or long-term continuous imaging [18,19]. The inherent flexibility of nanomaterials allows these devices to seamlessly integrate with the human body. It should be mentioned that the thermoacoustic effect should have a mechanical effect contribution at extremely high frequencies (such as applications in photoacoustic) [20,21], and this phenomenon lies outside the frequency range typically explored in current thermophone studies.

In these wearable applications, efficient acoustic focusing is essential for enhancing performance and effectiveness. For example, precise acoustic focusing can improve the resolution and sensitivity of ultrasound patches for medical imaging or targeted therapy. However, achieving such focusing with TA emitters presents significant challenges. Traditional geometric focusing methods impose stringent requirements on TA structure distribution and dimensions [22,23], while active control phased array focusing methods, though effective, are complex and difficult to implement [24,25]. Furthermore, while advancements in acoustic metamaterials enable highly effective sound concentration with minimal thickness, the intricate microstructures needed for precise beam focusing add significant fabrication complexity. These limitations restrict the broader application of TA emitters in wearable medical devices and other advanced technological fields [12].

To address these limitations, novel acoustic focusing techniques were extensively explored in the broader field of acoustics. Airy beams, known for their excellent focusing capabilities and properties such as abrupt focusing and self-healing, offer significant advantages in sound focusing [26]. The integration of acoustic metasurfaces and Airy beam focusing techniques provides a compact, simple, and efficient solution [27,28,29,30], where acoustic metasurfaces can be designed with only binary elements, significantly simplifying the fabrication process compared to traditional metasurfaces that require intricate microstructures. This combination allows for precise control of sound propagation and the realization of Airy beams, thus enhancing the focusing capabilities of TA emitters while reducing the manufacturing complexities.

Therefore, this study aims to investigate and improve the acoustic focusing capabilities and efficiency of TA emitters by integrating Airy beam and acoustic metasurface-based substrate. Specifically, it focuses on the following aspects: (1) evaluating the impact of metasurface-based substrates on the TA efficiency compared to traditional substrates, and (2) analyzing the impact of the integration of Airy beam-enhanced metasurface substrates with TA emitters on focusing intensity and the obstacle resistance. By addressing these aspects, this study provides a comprehensive solution to overcome the current limitations of TA emitters, thereby expanding their practical applications.

## 2. Integrated System Overview

In this section, we describe the overall structure of the integrated system, as shown in Figure 1a. The integrated system consists of two primary components: a thermoacoustic emitter and an acoustic metasurface. The TA emitter uses nanomaterials, such as carbon CNT films, to generate thermoacoustic waves through alternating current (AC) driving. This process involves periodic Joule heating, where heat is rapidly released into the surrounding fluent, causing thermal expansion and contraction. This results in periodic pressure waves, thereby generating sound. The acoustic metasurface in our system achieves sound focusing through phase modulation to form Airy beams.

The working principle of the integrated system is based on the interaction between the TA waves generated by the emitter and the acoustic metasurface. The TA emitter generates sound waves, which are then modulated by the acoustic metasurface to form Airy beams. This modulation allows for precise focusing of the sound waves. Figure 1a depicts the process of sound wave generation by the TA emitter, phase modulation by the metasurface, and the resultant focused sound wave.

The proposed integrated system offers significant advantages over traditional spherical geometric focusing TA technologies [17,22,23] or traditional metasurfaces, which are limited by size or complexity of intricate microstructures, respectively [25]. In these cases, acoustic intensity gradually accumulated in the non-focusing region before reaching the focal point, which leads to an undesirable off-target effect [28]. In contrast, our method uses a planar binary metasurface TA substrate, which eliminates dimensional constraints and reduces fabrication complexity of microstructures. Furthermore, the Airy beams provide superior focusing performance with a sharp accumulation at the focus and an abrupt increase in intensity, as shown in Figure 1b, by the controlled transverse self-acceleration and collapse of caustic at the focus in a nonlinear fashion [25,26,30]. These characteristics prevent off-target accumulation and ensure precise energy delivery [29]. The design of the acoustic metasurface and the explanation of how phase modulation is used to generate Airy beams are detailed in Section 4.

## 3. Thermoacoustic Emitter

In this section, we present the mathematical model of the thermoacoustic (TA) emitter. The generation of sound waves in a TA emitter can be described using the three equations of conservation of mass, momentum, and energy for the pressure and temperature in a fluid [31,32,33]:(1)$$\begin{array}{l} \frac{1}{B}\frac{\partial p}{\partial t}=\alpha_{T}\frac{\partial T}{\partial t}-\nabla \cdot v\\ \rho \frac{\partial v}{\partial t}=-\nabla p+\mu \nabla^{2}v+\left(\lambda +2\mu \right)\nabla \left(\nabla \cdot v\right)\\ \rho C_{p}\frac{\partial T}{\partial t}=\kappa \nabla^{2}T+\alpha_{T}T_{0}\frac{\partial p}{\partial t} \end{array}$$
where p is the fluid pressure, T and $T_{0}$ are the temperature and average ambient temperature, respectively, v is the particle velocity vector, ρ is the fluid density, B is the bulk modulus, $\alpha_{T}$ is the coefficient of volumetric expansion, λ and μ are the first and second viscosity coefficients, respectively, $C_{p}$ is the specific heat capacity at constant pressure, κ is the thermal conductivity. For an ideal fluid, the viscosity is zero, therefore: λ=μ=0 Equation (1) can be written as:(2)$$\begin{array}{l} \frac{1}{B}\frac{\partial p}{\partial t}=\alpha_{T}\frac{\partial T}{\partial t}-\nabla \cdot v\\ \rho \frac{\partial v}{\partial t}=-\nabla p\\ \rho C_{p}\frac{\partial T}{\partial t}=\kappa \nabla^{2}T+\alpha_{T}T_{0}\frac{\partial p}{\partial t} \end{array}$$
by eliminating v using the first two equations in Equation (2), the one-dimensional form of the thermoacoustic coupling control equation is obtained as:(3)$$\begin{array}{l} \frac{\partial^{2}p}{\partial t^{2}}-\frac{B}{\rho }\frac{\partial^{2}p}{\partial x^{2}}=\alpha_{T}B\frac{\partial^{2}T}{\partial t^{2}}\\ \frac{\partial T}{\partial t}-\frac{\kappa }{\rho C_{p}}\frac{\partial^{2}T}{\partial x^{2}}=\frac{\alpha_{T}T_{0}}{\rho C_{p}}\frac{\partial p}{\partial t} \end{array}$$
the equation can be used as the basic governing equation of thermoacoustic effect theory.

The input power is provided by an AC current applied to the thin film, generating periodic Joule heating. The heat source term Q can be defined as a sinusoidal function with frequency and amplitude ω/2 and I, respectively,
(4)$$Q={\left(I\sin\frac{1}{2}\omega t\right)}^{2}R=P_{in}\sin^{2}\frac{1}{2}\omega t$$ where R is the resistance of CNT film, $P_{in}=\left(I^{2}R\right)/2$ is the input power, and t is time. The heat source term Q can also be expressed in complex variables as
(5)$$Q=P_{in}\left(1-\cos\omega t\right)=P_{in}-P_{in}e^{i\omega t}$$
where $i=\sqrt{-1}$.

The temperature boundary conditions at the surface of film are defined as
(6)$$T\left(x=0\right)=T_{0}+T_{s}$$
(7)$$Q=2s\beta_{0}T_{s}+2sQ_{0}+sc_{s}\frac{dT_{s}}{dt}$$
where $\beta_{0}$ is the rate of heat loss per unit area, S is the single-side CNT film area, $c_{s}$ is the heat capacity per unit area, $T_{s}$ is the temperature above its surroundings, $Q_{0}=-\kappa \left(\partial T\left(x,t\right)/\partial x\right){|}_{x=0}$ is the instantaneous heat flow per unit area from CNT film to surrounding medium, and κ is the thermal conductivity of fluent.

For the initial analysis, we assume there is no reflection from the substrate, meaning the sound waves propagate freely without interference. The pressure boundary conditions at the surfaces are:(8)$$\frac{\partial p_{t}}{\partial n}=0$$
where n represents the normal at the boundary.

Without considering the substrate reflection, the analytical solution for the pressure response $p_{TA}$ can be derived by solving the wave equation with the given boundary conditions. The solution can be expressed as:(9)$$p_{TA}=\frac{p_{in}}{2s}\frac{\gamma -1}{\left(\beta_{0}+\kappa \sqrt{\frac{\omega }{2\alpha }}\right)+i\left(\kappa \sqrt{\frac{\omega }{2\alpha }}+\frac{1}{2}\omega c_{s}\right)}\sqrt{\frac{\omega }{\alpha }}\frac{\kappa }{c_{0}}e^{-i\left(\left(\omega /c_{0}\right)x+3\pi /4-\omega t\right)}.$$

The above mathematical model outlines the fundamental equations and boundary conditions governing the operation of the TA emitter. The solution derived here does not take substrate reflection into account. In the subsequent sections, we will consider the effects of reflections based on this initial pressure response, using it as the initial sound source. Furthermore, it is noticed that the mechanical effect [20,21] is not considered in this model since the frequency of the application scenario discussed in this paper is less than 1 MHz.

## 4. Formation of Autofocusing Quasi-Airy Beams with Metasurfaces

In this section, we discuss the design and function of acoustic metasurfaces for generating Airy beams. The study of Airy beams originated from quantum mechanics [34], initially as a solution to the Schrödinger equation and paraxial wave equation. The Airy function later found applications in optics [35] and acoustics [36]. 

The Airy equation(Stokes equation) is the simplest second-order linear differential equation. Its form is:(10)$$\frac{d^{2}y}{dx^{2}}=xy.$$

To solve this equation using the Laplace transform method, assume the solution to the equation is in the form:(11)$$y=\int_{c}e^{xt}v(t)dt.$$

Substituting the solution into the original equation yields:(12)$$\int_{c}t^{2}e^{xt}v(t)dt-\int_{c}xe^{xt}v(t)dt=0\to v^{\prime }\left(t\right)+t^{2}v\left(t\right)=0.$$

The solution is
(13)$$v\left(t\right)=e^{-\frac{t^{3}}{3}}.$$

Substituting this into Equation (11) yields two linearly independent solutions, one of which is the Airy function Ai(x), which can be expressed as:(14)$$\mathrm{Ai}(x)=\frac{1}{\pi }\int_{0}^{\infty }\cos(\frac{t^{3}}{3}+xt)dt.$$

The Airy beam theoretically possesses an infinite transverse size, making it impossible to realize in practical applications. Siviloglou [35] et al. modified the beam by “truncating” it, introducing an attenuation factor (α > 0, an exponential decay factor), thereby obtaining a finite-energy Airy solution at the initial plane. One of the remarkable properties of Airy beams is their non-diffracting nature, allowing them to propagate over long distances without significant changes in their width [30]. Furthermore, the radial symmetric Airy acoustic beam also exhibits self-healing and abrupt autofocusing properties, enabling the beam to concentrate its energy sharply at a focal point and reconstruct its trajectory after encountering an obstacle or perturbation [27]. Although the ideal Airy beam is a theoretical construct, our approach aims to approximate it in finite energy using a simple binary metasurface.

Considering the axisymmetric circular Airy acoustic beam distribution, the theoretical initial amplitude distribution of an axisymmetric distribution circular Airy beam in finite length can be described in a radial coordinate system by the Airy function:(15)$$p_{0}\left(r\right)=p_{TA}\times \mathrm{Ai}\left(\frac{r_{0}-r}{w}\right)\exp\left(\alpha \frac{r_{0}-r}{w}\right)$$
where $p_{TA}$ is the initial pressure amplitude of the TA emitter, which is obtained in Equation (9), Ai denotes the Airy function, $r_{0}$ is initial radial parameter which determined the position of Airy ring, *w* is a scaling factor that affects the size and spacing of the basement grooves of the metasurface, and α is a decay factor to ensure finite energy.

To achieve an ideal Airy beam, both the amplitude and phase of the wavefront must be controlled to match the desired distribution of *p*_0_ in Equation (15). However, controlling amplitude is complex and typically requires intricate microstructures within the metasurface. In contrast, phase control is comparatively simpler. Previous studies [28,29,30] showed that by modulating only the phase to match the required phase distribution of the *p*_0_, while maintaining a constant amplitude, the resulting approximate initial pressure distribution *p*_1_ can still exhibit excellent focusing properties. Although this approximation means that *p*_1_ does not fully meet the amplitude requirements of an Airy beam distribution, phase control alone significantly reduces implementation complexity.

To realize the quasi-Airy beam initial pressure distribution *p*_1_, an acoustic metasurface is used to modulate the phase of the incident sound waves generated by the TA emitter. Figure 1c–e illustrates the conceptual design of the metasurface and the resultant approximate Airy beam. The metasurface is designed with a planar configuration incorporating binary elements that induce binary phase shifts of 0 and π to the reflected waves. This is achieved by structuring the surface with grooves of λ/4 depth, that create a path difference equivalent to a half wavelength. This phase modulation aligns the reflected wavefronts to form the quasi-Airy beam, maintaining the essential properties of self-healing and abrupt autofocusing [37,38]. It is worth mentioning that this radially symmetric quasi-Airy beam retains some of the non-diffracting properties, but it lacks the self-accelerating characteristics of one-dimensional Airy beams [39]. These features enable the beam to concentrate energy sharply at a focal point and recover its trajectory after encountering obstacles.

## 5. Results and Discussion

By focusing on phase modulation rather than amplitude control, the metasurface design simplifies fabrication while still approximating the ideal Airy beam. This approach retains the essential functional characteristics needed for effective beam formation and control.

This section systematically examines several key aspects of the TA-integrated system’s performance, including heat conduction, acoustic focusing, and the self-healing properties of the quasi-Airy beam. Simulations were conducted using finite element analysis within the thermoacoustic framework, with water as the medium. The substrate material was chosen as polyimide (PI) [30], with a density of 1423 kg/m^3^, sound speed of 2212 m/s, specific heat capacity at a constant pressure of 1100 J/(kg·K), and thermal conductivity of 0.15 W/(m·K). CNT films were selected as the TA emitter due to their favorable properties [17], including density of 472 kg/m^3^, sound speed of 1500 m/s, specific heat capacity of 716 J/(kg·K), and thermal conductivity of 0.6 W/(m·K). The focusing performance was characterized by the intensity contrast ratio ***G***, defined as the ratio of the acoustic intensity at each point in the computational domain to the average intensity of the initial thermoacoustic source.

### 5.1. Heat Conduction and Acoustic Performance Analysis of TA-Integrated System

Understanding of the heat conduction mechanisms is essential for optimizing the TA-integrated system’s efficiency in generating and focusing acoustic waves. This section presents analysis of the heat conduction behavior between the TA emitter, the metasurface substrate, and the intervening gap. 

Figure 2a illustrates the simplified schematic about heat conduction within a specific region of the TA-integrated system. This schematic highlights the thin-film TA emitter and a metasurface substrate, separated by a small gap. The thermal diffusion length, which represents the maximum distance heat can propagate through the medium, is crucial for the efficient generation and propagation of thermoacoustic waves. The thermal diffusion length *L_d_* is given by:(16)$$L_{d}=2\pi \sqrt{\frac{\kappa }{\rho C_{p}\pi \omega }}$$
where κ is the thermal conductivity, ρ is the density, $c_{p}$ is the specific heat capacity, and ω is the angular frequency of the thermal wave.

Figure 2b depicts the relationship between the temperature rise of the TA emitter and time under alternating current, where the temperature follows a sinusoidal pattern with a period half that of the current. Figure 2c further explores the relationship between the amplitude of the temperature rise/SPLs and the heat capacity per unit area, highlighting the critical role of the TA emitter’s surface heat capacity in determining both heat conduction and TA efficiency within the system. This explains why CNT films are excellent candidates for TA emitters, as they possess a low heat capacity.

Figure 2d shows how the thermal diffusion length varies with frequency, indicating that higher frequencies lead to faster thermal attenuation. Figure 2e provides a detailed curve of temperature variation with distance, with an inset comparing the thermal diffusion lengths in water and the substrate. The analysis suggests that maintaining an appropriate gap can enhance heat transfer and improve TA generation performance within the system.

Figure 2f presents the distribution of sound intensity along the central axis under different heating power levels, while Figure 2g further details the acoustic intensity and intensity contrast ratio *G* specifically at the focal point. As heating power increases, the sound intensity follows a parabolic growth pattern, while the intensity contrast ratio *G* remains stable. This suggests that the system can effectively enhance acoustic output by adjusting heating power without compromising focal stability.

In summary, the heat conduction analysis of the TA-integrated system highlights the key factors that influence the generation and propagation of thermoacoustic waves. The system’s performance is significantly impacted by the thermal diffusion length, the heat capacity of the TA emitter, and the applied heating power.

### 5.2. Parameters Design and Sub-Wavelength Focusing of TA-Integrated System

In this study, the focusing performance of a TA-integrated system by systematically varying the initial radius (*r*_0_) and scaling factor (*w*) is investigated.

The parametric analysis results demonstrate that the appropriate selection of *r_0_* and *w* can significantly enhance the intensity contrast ratio *G*. Specifically, the maximum intensity contrast ratio was achieved at 1 MHz when *r*_0_ = 6.6 λ and *w* = 1.2 λ, as shown in Figure 3a. Furthermore, the focal length *Z*_focus_ was found to vary substantially with changes in r0 and w, ranging from 15.2λ to 61λ, as depicted in Figure 3b. This highlights the versatility of the metasurface design in achieving precise focal positioning.

In addition to the focal length, the radial full width at half-maximum (FWHM) of the focal spot was examined across the parameter space, as shown in Figure 3c. The results indicate that sub-wavelength focusing can be stably achieved, with the maximum radial FWHM being 0.65λ. The axial full length at half-maximum (FLHM) was also assessed, with the minimum value reaching 1λ, as illustrated in Figure 3d.

These simulation results clearly validate the effectiveness and feasibility of the binary-coded TA metasurface reflector in achieving sub-wavelength ultrasound focusing and precise axial focal positioning.

### 5.3. Frequency-Controlled Focal Length Tunability of TA-Integrated System

To ensure optimal focusing performance at 1 MHz, the metasurface was designed with an initial radius *r*_0_ = 6.6 λ and a scaling factor w = 1.2 λ, as determined in the previous section. We then investigated the influence of frequency on the tunability of the focal length in the TA-integrated system. The objective was to understand how varying the operational frequency impacts the focal position, thereby enabling precise control over the focus without altering the physical design of the TA-integrated system.

As shown in Figure 4a, the intensity contrast ratio *G* remains consistently above 1000 across a broad frequency range from 0.88 MHz to 1.09 MHz, demonstrating the TA-integrated system’s capability for robust broadband focusing. Furthermore, as illustrated in Figure 4b, the focal length Z focus was found to increase linearly with frequency, highlighting the system’s effective tunability. This linear relationship enables precise control of the focal position solely through frequency adjustment, without the need to modify the physical structure of the metasurface.

Sub-wavelength focusing was consistently maintained across the frequency range of 0.89 MHz to 1.09 MHz, with the radial FWHM remaining below 0.6λ, as shown in Figure 4c. This highlights the system’s ability to concentrate acoustic energy within a sub-wavelength region, which is crucial for high-resolution applications. However, a trade-off was observed in the axial direction. Figure 4d shows that the axial FLHM increases with frequency, indicating a broadening of the focal region along the *z*-axis. This suggests that while radial focusing remains sharp, the focal depth extends at higher frequencies. Figure 4e illustrates the normalized acoustic intensity distributions at working frequencies of 0.88 MHz, 0.94 MHz, 1.00 MHz, and 1.06 MHz. These results confirm that as the focal point shifts along the axial direction with increasing frequency, the system consistently maintains sharp radial focusing, with the FWHM remaining stable around 0.6λ.

In summary, the designed TA-integrated system demonstrates both frequency-dependent focal length control and robust sub-wavelength focusing across a broad frequency range, making it a versatile tool for applications requiring precise acoustic control.

### 5.4. Self-Healing Capability of the Focused Quasi-Airy Beams

In practical applications, particularly in medical ultrasound, the ability of an acoustic system to maintain focused beams in the presence of obstacles is critical. This section investigates the self-healing properties of the focused quasi-Airy beams generated by the TA-integrated system, specifically its ability to preserve focal sharpness and intensity when encountering scattering objects. To evaluate this, numerical simulations were performed while the obstacles were modeled with material properties similar to human rib bones, as depicted in Figure 5.

Figure 5b–d illustrates the effects of different obstacle shapes placed at (r,z) = (15λ,15λ) on the intensity contrast ratio *G* at the focal point. The results show that although *G* decreases in the presence of obstacles compared to the obstacle-free case, the focal length, FLHM, and FWHM remain relatively stable. This indicates that the quasi-Airy beams possess inherent self-healing properties, enabling it to maintain focusing performance despite the presence of scattering objects.

Figure 5e–h further analyzes the impact of obstacle size on the self-healing capability. As the obstacle size increases, the intensity contrast ratio *G* at the focal point decreases. At the same time, the axial FLHM increases, while the radial FWHM consistently remains at a sub-wavelength level of approximately 0.5λ. This suggests that the TA-integrated system can reliably maintain sharp focusing and effective operation even in challenging environments with varying types of physical barriers.

Considering the potential application of the TA-integrated system in medical ultrasound, its self-healing capability was tested with array-like obstacles, as shown in Figure 5i. These obstacles model a lung ultrasound scenario, with radius sizes of 1.5 mm and 3 mm. The results show that even with such arrays, the system maintains robust focusing performance, with the focal position remaining stable. The system consistently achieves acoustic focusing across a broad frequency range, with the focal length increasing as the operating frequency rises. This robust self-healing performance highlights the TA-integrated system’s potential value in medical ultrasound applications, where maintaining focus in the presence of complex anatomical structures is crucial.

In conclusion, this section systematically examined the key factors affecting the TA-integrated system, focusing on heat conduction, acoustic focusing, and self-healing properties. The analyses in Section 5.1, Section 5.2, Section 5.3 and Section 5.4 reveal that parameters such as thermal diffusion length, surface heat capacity, and operating frequency critically influence the system’s ability to generate and focus thermoacoustic waves. The robustness of the system under different obstacle scenarios highlights its potential for stable performance in practical applications. These findings offer insight into the underlying mechanisms, setting the stage for further refinements and optimizations of the TA-integrated system.

## 6. Conclusions

This study represents a significant advancement in the TA focusing system by integrating Airy beam-enhanced acoustic metasurfaces, addressing challenges in precise acoustic focusing and efficient energy conversion. The system shows significant improvements in sub-wavelength focusing and tunable focal length, even when obstacles are present. These benefits are due to the unique properties of quasi-Airy beams, which offer better focusing and self-healing. The reduced fabrication complexity further supports practical use across various TA applications. The findings suggest strong potential for applications in medical ultrasound, as well as other fields, such as wearable devices and non-invasive therapies. Future work will optimize the system for specific medical uses, such as high-resolution imaging and targeted therapy, and explore metasurface designs in other TA systems.

## Figures and Tables

**Figure 1 nanomaterials-14-01481-f001:**
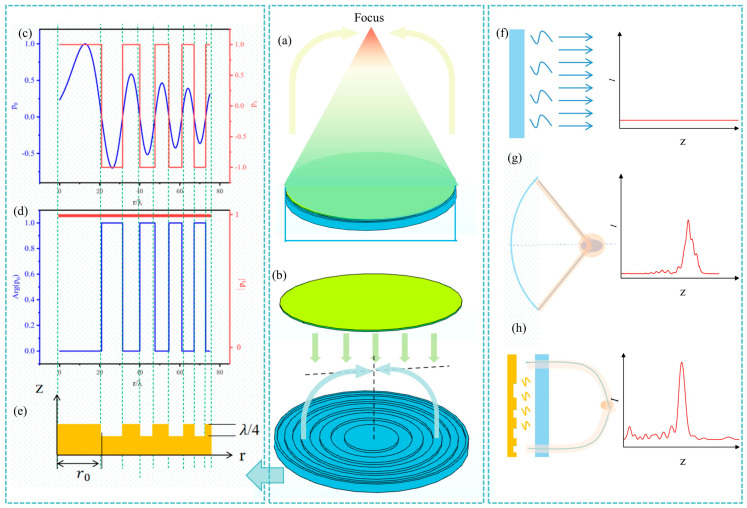
Overview and phase modulation of the integrated thermoacoustic (TA) focusing system. (**a**) Schematic illustration of the integrated TA focusing system; (**b**) structure of the integrated system consisting of a TA emitter and a metasurface substrate; (**c**) pressure distribution *p*_0_ (blue line) of an ideal Airy beam and the approximate distribution p1 (red line); (**d**) phase and amplitude distribution of p1; (**e**) metasurface structure enabling phase-only modulation to approximating the quasi-Airy beam profile; (**f**–**h**) and an illustration of thermoacoustic wave propagation on a planar surface, a traditional spherical surface, and the integrated TA focusing system.

**Figure 2 nanomaterials-14-01481-f002:**
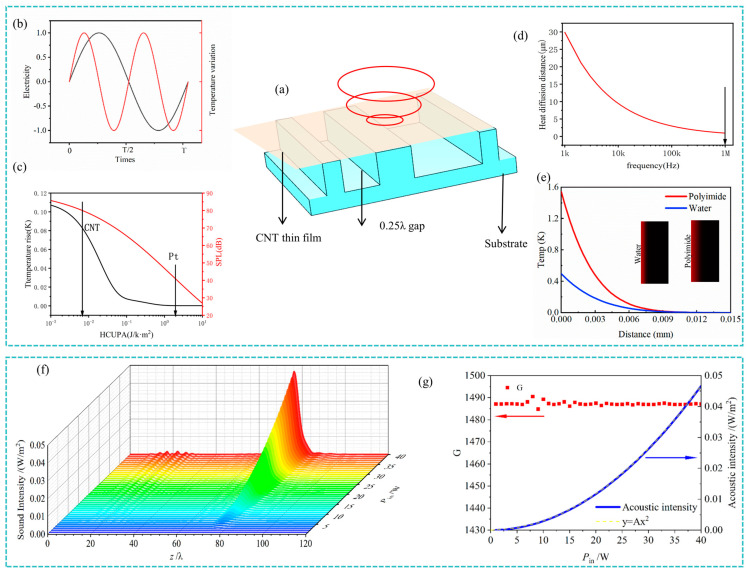
Heat conduction and acoustic performance analysis of the integrated thermoacoustic (TA) system: (**a**) a section of the integrated TA focusing system; (**b**) the excitation current (black line) and temperature rise (red line) of the TA emitter; (**c**) variation in temperature rise (black line) and SPL (red line) with the material properties of film; (**d**) variation in thermal diffusion length with frequency; (**e**) temperature decay with distance, with the inset comparing thermal diffusion lengths in gap and the substrate; (**f**) sound intensity along the central axis (*r* = 0) under different heating powers; and (**g**) intensity contrast ratio *G* and acoustic intensity at the focal point as functions of input power.

**Figure 3 nanomaterials-14-01481-f003:**
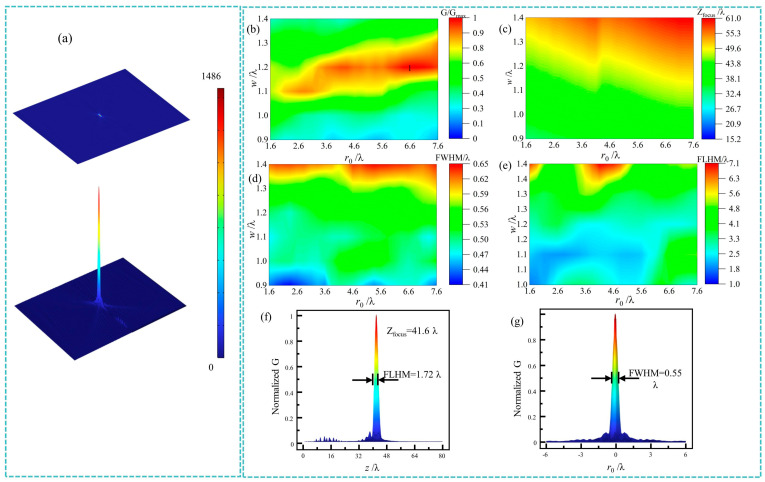
Parameters analysis (*r*_0_, *w*) of the metasurface in the integrated thermoacoustic (TA) focusing system; (**a**) the sound intensity contrast ratio *G* at 1 MHz (*r*_0_ = 6.6 λ, *w* = 1.2 λ); (**b**–**e**) variation in the normalized intensity contrast ratio *G*, focal length *Z_focus_*, full width at half-maximum (FWHM), and full length at half-maximum (FLHM) with the initial radial parameter *r*_0_ and scaling factor *w* of the metasurface. (**f**,**g**) Normalized intensity contrast ratio *G* along the axial direction (*r* = 0) and along the radial direction (*Z* = 41.6 λ).

**Figure 4 nanomaterials-14-01481-f004:**
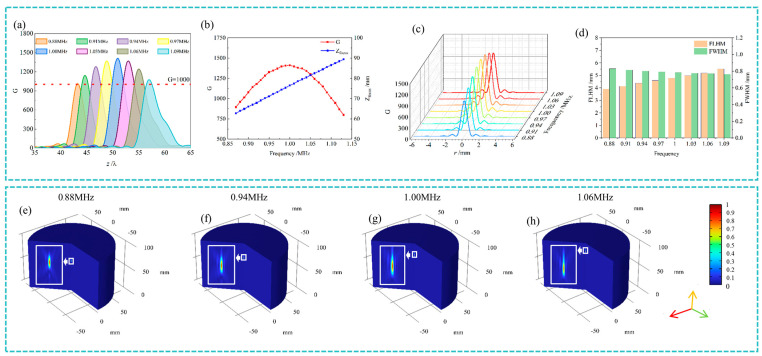
Impact of frequency on focusing characteristics in the thermoacoustic (TA)-integrated system: (**a**,**b**) intensity contrast ratio *G* and focal length *Z*_focus_ across a broad frequency range (0.88 MHz to 1.09 MHz); (**c**,**d**) variation in the full width at half-maximum (FWHM) and full length at half-maximum (FLHM) with the frequency; and (**e**–**h**) normalized acoustic intensity distributions at frequencies of 0.88 MHz, 0.94 MHz, 1.00 MHz, and 1.06 MHz.

**Figure 5 nanomaterials-14-01481-f005:**
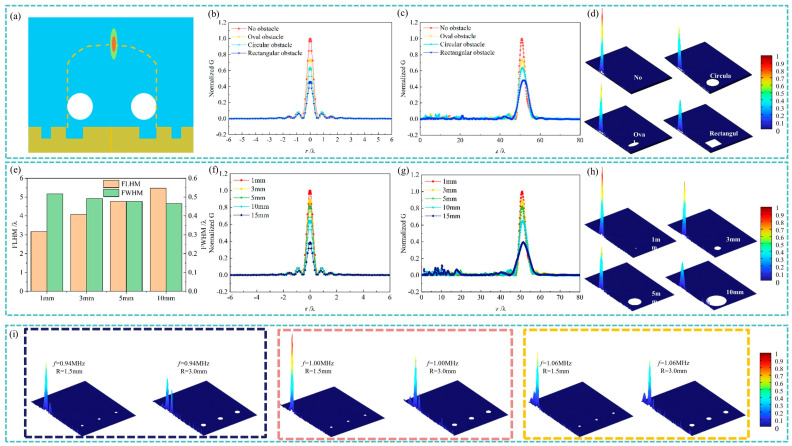
Self-healing focusing of the thermoacoustic (TA)-integrated system in the presence of obstacles: (**a**) schematic illustration of the obstacle placement; (**b**–**d**) effects of different obstacle shapes (oval, circular, and rectangular) on the intensity contrast ratio *G* at the focal point; (**e**–**h**) impact of obstacle size on the self-healing capability of the system; and (**i**) self-healing focusing in the presence of array-like obstacles.

## Data Availability

Data are contained within the article.

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
