# Peer review of "Simulation Analysis of Thermoacoustic Effect of CNT Film with Metasurface-Enhanced Acoustic Autofocusing"

_nanomaterials, 2024, doi:10.3390/nano14181481_

Round 1

Reviewer 1 Report

Comments and Suggestions for Authors

Comments are collected in the attached file.

Reviewer 2 Report

Comments and Suggestions for Authors

The manuscript describes a thermoacoustic system composed of two main parts: (i) a generation layer where the thermoacoustic effect is active as in a classical thermopone (here made by carbon nanotube CNT), and (ii) a focusing system integrating Airy beams and acoustic metasurfaces. The idea of using a focusing system coupled with a thermoacoustic generator is original and could be of interest for biomedical applications (both diagnostic and therapy). The paper is quite clear and deserves publication after the improvement of the following points:

1) The main equations of the classical thermoacoustic generation are stated in Eq.1 and 2 without discussion. It should be clarified if these equations are fully correct, thus implementing all the physical phenomena, or they consider some approximation. To do this, the authors should mention and compare the equations with the manuscript: Pierre Guiraud, Stefano Giordano, Olivier Bou-Matar, Philippe Pernod, Raphael Lardat, "Multilayer modeling of thermoacoustic sound generation for thermophone analysis and design", Journal of Sound and Vibration 455, 275-298 (2019), DOI: 10.1016/j.jsv.2019.05.001. The limitations of the adopted approach should be carefully discussd to help the readers the better appreciate the potentialities of the method and its validity.

2) It has been recently proved that the thermoacoustic effect is based on the thermophone mechanism when the frequency is low-medium, and on the mechaniphone effect when the frequancyis medium-high. See for instance: Michele Diego, Marco Gandolfi, Stefano Giordano, Fabien Vialla, Aurélien Crut, Fabrice Vallée, Paolo Maioli, Natalia Del Fatti, and Francesco Banfi, "Tuning photoacoustics with nanotransducers via thermal boundary resistance and laser pulse duration", Appl. Phys. Lett. 121, 252201 (2022), DOI: 10.1063/5.0135147; and: Stefano Giordano, Michele Diego, and Francesco Banfi, "Acoustic Wave Generation in Nanofluids: Effect of the Kapitza Resistance on the Thermophone to Mechanophone Generation Mechanism Transition", J. Phys. Chem. C 127, 10227−10244 (2023), DOI: 10.1021/acs.jpcc.3c01808.

It means that for frequencies larger than a given threshold, the vibrations of the generation layer become important and cannot be neglected (mechanophone effect). This must be taken into account by considering the elasticity of the generation layer, a point neglected in this work. However, the problem should be mentioned and discussed for possible future generalizations and to better describe the limits of validity of the presented scheme.

3) Although quasi-airy beams with metasurfaces are known objects in advanced acoustics, it could be useful to introduce these systems with a larger discussion better explaining their origin and their physical behavior. This should be very useful for the reader. For example, Eq.9 is written wthout any discussion about its derivation, its limits of validity, etc.  This point should be carefully improved.

Round 2

Reviewer 1 Report

Comments and Suggestions for Authors

The authors took my comments into account.

Reviewer 2 Report

Comments and Suggestions for Authors

The authors have taken into considerations all the suggestions proposed by the referee and modified the manuscript accordingly. The paper can be now accepted for publication in opinion of the present reviewer.